# Time-resolved THz Stark spectroscopy of molecules in solution

Bong Joo Kang[1,2], Egmont J. Rohwer[1], David Rohrbach [1], Elnaz Zyaee [1], Maryam Akbarimoosavi[1], Zoltan Ollmann[1], Gleb Sorohhov[3], Alex Borgoo [4], Michele Cascella [4], Andrea Cannizzo[1], Silvio Decurtins[3], Robert J. Stanley [5], Shi-Xia Liu [3] & Thomas Feurer[1]✉

For decades, it was considered all but impossible to perform Stark spectroscopy on molecules in a liquid solution, because their concomitant orientation to the applied electric field results in overwhelming background signals. A way out was to immobilize the solute molecules by freezing the solvent. While mitigating solute orientation, freezing removes the possibility to study molecules in liquid environments at ambient conditions. Here we demonstrate time-resolved THz Stark spectroscopy, utilizing intense single-cycle terahertz pulses as electric field source. At THz frequencies, solute molecules have no time to orient their dipole moments. Hence, dynamic Stark spectroscopy on the time scales of molecular vibrations or rotations in both non-polar and polar solvents at arbitrary temperatures is now possible. We verify THz Stark spectroscopy for two judiciously selected molecular systems and compare the results to conventional Stark spectroscopy and first principle calculations.

Stark spectroscopy is an invaluable tool to reveal information about physicochemical properties of molecules[1–9]. Sufficiently strong electric fields modify absorption spectra of isotropic ensembles of molecules if ground and excited state energy eigenvalues of the respective optical transition shift in energy due to the interaction with the applied electric field. In most cases, the interaction can be treated as a perturbation and expanded in a power series of the electric field. In first order, Stark spectroscopy reveals information on the electric dipoles, and in the second order, on the induced dipoles, i.e., on the polarizabilities of the two states. The former is referred to as the linear Stark effect (Fig. 1a), the latter is called the quadratic Stark effect (Fig. 1b). It is important to note that only a nonzero difference between ground and excited state dipole or polarizability results in a modified absorption spectrum, and consequently, only those differences can be extracted from a measurement. While a change in dipole moment, for instance, reflects the degree of charge separation or charge transfer associated with the transition, a change in polarizability describes the sensitivity of a transition to an external electric field[2]. These effects are also known as

electro-chromism[5] and a corresponding measurement gives insight in, for instance, photo-induced electron or charge transfer[10–14], nonlinear material properties[2,4,8,15], biological organization and energy tuning[16–21], or solvato-chromism. Also, vibrational Stark spectroscopy of CO or CN ligands has developed into an important tool to measure in situ electric field strength in various chemical environments[22]. In addition, more recent research results revived the interest in the observation of transient Stark responses in solids, gases, or other media such as polymers, quantum dots, and TMDC[23–27]. However, transient Stark spectroscopy of molecules in solution at room temperature has not yet been demonstrated, because a successful measurement of molecules in solution is subject to some preconditions.

Conventional Stark spectroscopy uses low frequency (kHz) electric fields, which oscillate much slower than typical rotation times of molecules in solution. Hence, molecules must be immobilized in order to avoid alignment of their dipoles along the applied electric field, otherwise this poling effect would result in an overwhelming increase of the overall absorption and obscure any Stark signatures. Typically,

[1]Institute of Applied Physics, University of Bern, Bern, Switzerland. [2]Division of Advanced Materials, Korea Research Institute of Chemical Technology (KRICT), Daejeon, Republic of Korea. [3]Department of Chemistry, Biochemistry and Pharmaceutical Sciences, University of Bern, Bern, Switzerland. [4]Department of Chemistry and Hylleraas Centre for Quantum Molecular Sciences, University of Oslo, Oslo, Norway. [5]Department of Chemistry, Temple University, Philadelphia, PA, USA. ✉e-mail: thomas.feurer@unibe.ch

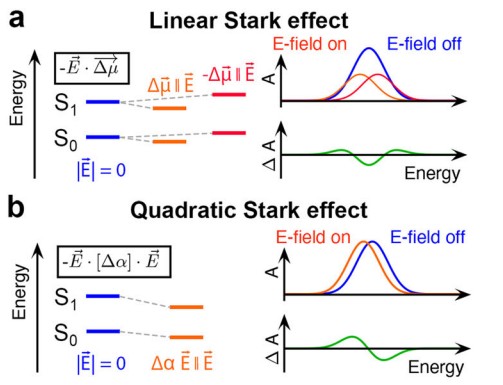

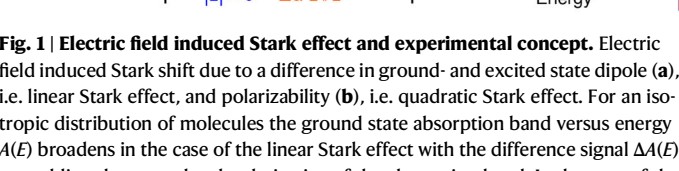

**Fig. 1 | Electric field induced Stark effect and experimental concept.** Electric field induced Stark shift due to a difference in ground- and excited state dipole (**a**), i.e. linear Stark effect, and polarizability (**b**), i.e. quadratic Stark effect. For an isotropic distribution of molecules the ground state absorption band versus energy $A(E)$ broadens in the case of the linear Stark effect with the difference signal $\Delta A(E)$ resembling the second order derivative of the absorption band. In the case of the quadratic Stark effect the ground state absorption band shifts resulting in a difference signal $\Delta A(E)$ that is proportional to the first order derivative of absorption band. **c** Schematic representation of the experimental setup. The femtosecond supercontinuum probe pulse is scanned in time across the collinear single-cycle THz pulses and its spectrum is recorded by a spectrometer.

this is achieved by freezing the solvent, which limits the range of solvents that can be used since these need to form optical glasses to avoid scattering of probe light. Moreover, freezing prevents molecules from being characterized in their natural, liquid, or physiological environment. Finally, the sample geometry, with the electrodes on the front- and backside of the sample cuvette, leads to a non-optimal geometry with the angle between the polarization of the Stark field and the probe being far from the ideal value of 0 deg (typically 50 to 60 deg)[28].

Here, we demonstrate that increasing the frequency of the oscillating electric field to the terahertz (THz) regime removes all of the above-mentioned constraints and disadvantages of conventional Stark spectroscopy. Over a number of years, it became possible to generate phase stable single- or few-cycle THz pulses with sufficiently strong electric fields (up to or even exceeding 1 MV/cm) to induce measurable Stark shifts as well as femtosecond supercontinuum (fs-SC) probe pulses to time-resolve those transient Stark signatures[29,30]. A schematic of the experimental realization is shown in Fig. 1c. Scanning the time delay between probe pulse and the THz waveform allows for the measurement of Stark signatures for positive or negative electric fields and from zero field up to the peak field strength of the THz pulse (see Supplementary Information 4). At THz frequencies the electric field oscillates faster than typical molecular rotation times in solution, hence for the first time, time-resolved Stark spectroscopy of molecules is performed without freezing the sample. Consequently, virtually any solvent can be used and molecules can be studied in their natural chemical or biological environment. Moreover, a wider temperature range becomes accessible, especially temperatures above the freezing point of the solvent but also room temperature or higher. The THz frequency of the field source also entails a higher dielectric breakdown allowing for electric field strengths, which were impossible to attain previously[31–33]. At such field strengths, for instance, transition polarizability or hyperpolarizability may play a more important role, particularly for weakly allowed transitions. Such higher-order electric field effects may require a new theoretical concept, especially when the applied electric field can no longer be treated as a perturbation to the system's Hamiltonian or when magnetic effects need to be considered. THz Stark spectroscopy offers additional minor advantages, i.e., the sample thickness can be substantially increased and is limited only by absorption or velocity matching between the THz and the probe pulse. Additionally, arbitrary angles between the electric field vector and the probe polarization can be used and the absence of electrodes avoids potential redox chemistry in the pristine sample.

For the present THz Stark spectroscopic study, we selected two molecules to separately study the dynamics for linear and quadratic Stark effect and for comparison we performed quantum chemical calculations as well as conventional Stark spectroscopy. One molecule consists of a strong electron donor tetrathiafulvalene and an electron acceptor benzothiadiazole, showing an energetically low-lying intramolecular charge transfer state with a substantial change in dipole moment. The other is an anthanthrene derivative tetrasubstituted with silyl-protected acetylene groups to extend its $\pi$-conjugation, leading to an intense and sharp absorption band with a large change in polarizability.

## Results and discussion
### Dynamics of the Stark signature
We first discuss the experimental results of a molecule with a pronounced intramolecular charge-transfer (ICT) character, namely an annulated electron donor-acceptor compound. Thereby, tetrathiafulvalene (TTF) acts as a strong donor and benzothiadiazole (BTD) equally as an acceptor within the compact and planar dyad[34,35]. The bromine functionalization of BTD increases its acceptor strength and the propyl chains on TTF act as solubilizing groups (Fig. 2d) (see Supplementary Information 1). Figure 2a shows the color-coded THz-induced change in absorption $\Delta A(\tau, \lambda)$ in TTF-BTD versus time delay $\tau$ and wavelength $\lambda$ for parallel polarization orientation between probe pulse and THz waveform (corresponding results for a perpendicular orientation are provided in the Supplementary Information 5). The color scale indicates the difference in absorption in units of optical density (OD). The two-dimensional distribution reveals both temporal and spectral characteristics of the THz field-induced Stark shift with a time and wavelength resolution of approximately 100 fs and 0.5 nm. Time zero is set arbitrarily to the maximum change in absorption $\Delta A_{max}$. The two peaks of the single-cycle THz pulse are well resolved with zero signal at the zero-crossing of the THz electric field. The measured spectra around the peaks of the THz field indicate a spectral broadening of the rather broad $S_0 - S_1$ absorption band between 390 nm and 650 nm, which is not unusual for ICT transitions.

Calculating the margin along the delay axis between the two green dotted vertical lines results in the average Stark signal versus wavelength (green curve in Fig. 2b). Note that all measurements had the background signal (from pure toluene in the cuvette) subtracted. Comparing the measured signal to the scaled first- and second-order derivative of the ground state absorption curve reveals that the Stark signature is dominated by a first-order Stark contribution (black

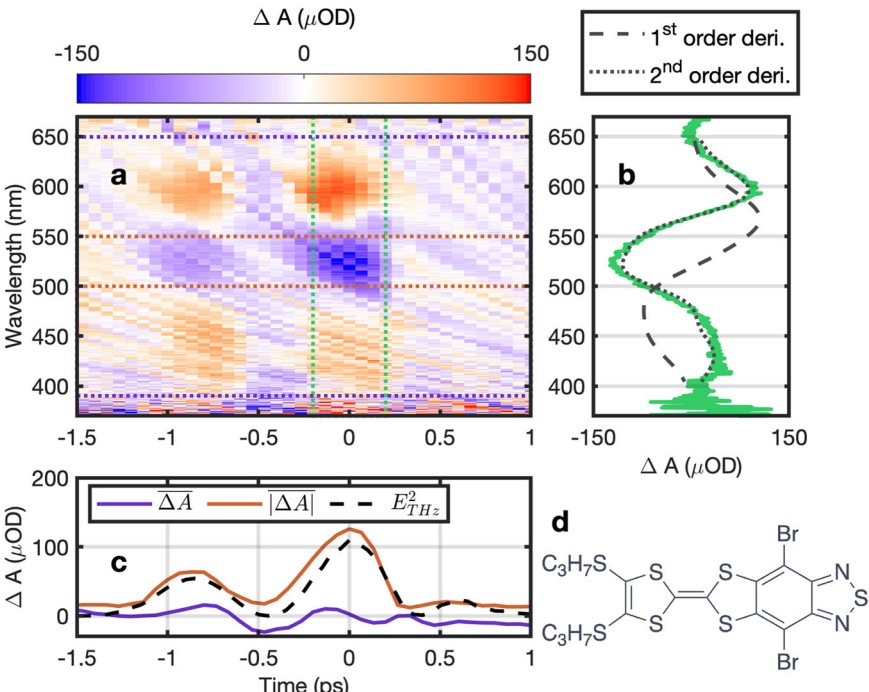

**Fig. 2 | THz Stark signal of TTF-BTD with parallel orientation. a** False-color plot of the measured change in absorption spectrum as a function of time delay between THz and probe pulse and wavelength. **b** Time-averaged (between the two green dotted lines) change in absorption versus wavelength (green solid curve) compared to the scaled first (black dashed curve) and second order derivative (black dotted curve) of the ground state absorption spectrum. **c** Spectral average of the change in absorption between the purple dotted (purple curve) and red dotted lines (red curve) in (**a**). The red curve is compared to the scaled square of the measured THz electric field $E^2_{\text{THz}}$ (black dashed curve). **d** Chemical structure of TTF-BTD.

dotted curve) revealing a linear Stark effect caused by a change in dipole moment between ground and excited state. Additionally, we calculate two margins along the wavelength axis, first between the purple dotted horizontal lines resulting in the purple curve in (Fig. 2c), and second between the two red dotted horizontal lines yielding the red curve in (Fig. 2c). The margin over the entire transition (purple curve) versus time delay is essentially zero confirming that no alignment of molecules to the applied THz field occurs. The residual small signal is most likely due to an imperfect correction of the group velocity dispersion in the fs-SC (see Supplementary Information 4) as well as a sensitivity level of 20 $\mu$OD due to fs-SC fluctuations. These findings constitute an important result as they demonstrate that THz Stark spectroscopy can be applied to molecules in solution without the need to freeze the solvent. The red curve, averaged over one of the peaks in the Stark spectrum, indicates that the instantaneous Stark signal scales as the square of the THz field which is in agreement with Liptay's derivation of the linear and the quadratic Stark effect of an ensemble of molecules with isotropic orientation.

The second molecule, anthanthrene[36], is a $\pi$-conjugated organic molecule of interest due to its semiconducting properties and potential applications in light emitting diodes or solar cells. The structure of anthanthrene is shown in Fig. 3d. It constitutes an interesting building block for organic electronics and Stark spectroscopy has the potential to reveal some of its relevant physicochemical properties. The transient Stark signal of the $S_0 - S_1$ transition of anthanthrene in toluene is shown in Fig. 3. The polarization between probe pulse and the THz waveform was parallel and the corresponding results for perpendicular polarization are provided in the Supplementary Information 5. The electronic transition shows a well separated vibrational progression, hence all vibrational bands are treated as one transition within our detection window. Margins are calculated following the same recipe as outlined above. A comparison of the time-averaged Stark signature (Fig. 3b green curve) with the scaled first-

(black dashed curve) and second-order derivative (black dotted curve) of the ground state absorption reveals a mostly quadratic Stark effect related to a difference in polarizability of ground and excited state. It also suggests that ground and excited state dipoles are, very likely based on symmetry arguments, negligible or similar in value. The two margins along the wavelength axis shown in Fig. 3c (purple and red curves) confirm that poling effects can be excluded, hence the sample maintains an isotropic distribution throughout, confirming again that the THz field oscillation is faster than the rotation time of investigated molecule in solution.

That is, both molecules show a pronounced instantaneous Stark signature, which is well-matched to either the first- or the second-order derivative of the ground state absorption band, suggesting a dominant quadratic or linear Stark effect, respectively. In both cases, the Stark signal is proportional to the square of the THz electric field in agreement with Liptay formalism (see Supplementary Information 7) tracing the picosecond single-cycle THz waveform and we identify no measurable hysteresis or memory effect.

**Comparison between conventional and THz Stark spectroscopy**

For both molecules, the Stark spectra for parallel and perpendicular polarization between probe pulse and THz waveform are fitted simultaneously and analyzed using the formalism outlined by Liptay[5,37] to extract relevant molecular parameters, such as differences in dipole moment or polarizability. We compare the results to those obtained from conventional Stark spectroscopy (see Supplementary Information 3) and those calculated from Density Functional Theory (DFT). DFT also provides molecular orbitals and associated energies for further discussion of results, specifically the charge redistribution associated with an excitation (see Supplementary Information 2). Recall that conventional Stark spectroscopy is done at 77 K, that is well below the freezing temperature of toluene, whereas THz Stark spectroscopy can be performed in principle at any temperature, but here is done at

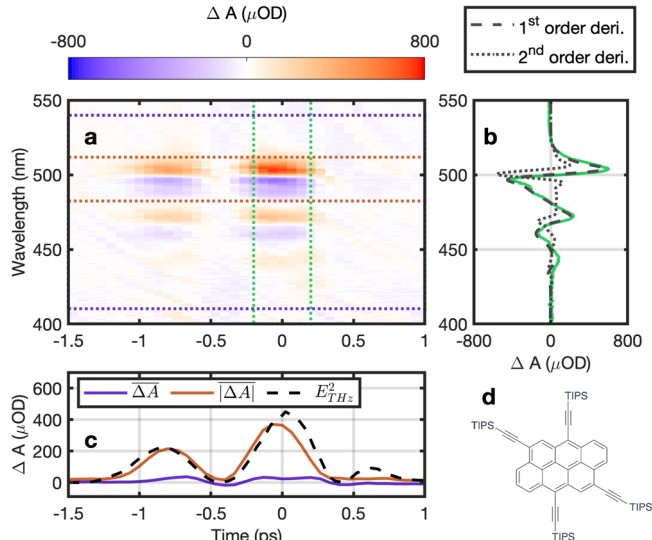

**Fig. 3 | THz Stark signal of anthanthrene with parallel orientation. a** False-color plot of the measured change in absorption spectrum as a function of time delay between THz and probe pulse and wavelength. **b** Time-averaged (between the two green dotted lines) change in absorption versus wavelength (green solid curve) compared to the scaled first (black dashed curve) and second order derivative (black dotted curve) of the ground state absorption spectrum. **c** Spectral average of the change in absorption between the purple dotted (purple curve) and red dotted lines (red curve) in (**a**). The red curve is compared to the scaled square of the measured THz electric field $E^2_{\mathrm{THz}}$ (black dashed curve). **d** Chemical structure of anthanthrene.

room temperature. To account for different optical path lengths, sample concentrations, or electric field strength in the two measurement techniques, we compare the change in molar attenuation coefficient $\Delta\epsilon$ scaled to 1 MV/cm rather than the change in absorption.

The ground state absorption spectra of TTF-BTD and anthanthrene in toluene are shown in Fig. 4a and b for 77 K (blue curves) and room temperature (red curves). The DFT calculations suggest that TTF-BTD exhibits a broad absorption band due to a substantial ICT in the excited state. Anthanthrene, on the other hand, exhibits relatively narrow absorption features with an evident vibronic progression and DFT calculations mainly suggest a change in polarizability. When decreasing the temperature from 300 K to 77 K toluene is known to increase its effective polarity due to an increase in density[38]. While in TTF-BTD this results in a blue-shift of the absorption peak from 514 nm to 505 nm due to the solvent's instantaneous electronic polarizability[39], in anthanthrene this leads to increased stabilization of the energy levels and an associated red-shift of the lowest vibronic mode of the $S_0 - S_1$ HOMO - LUMO transition from 500 nm to 505 nm. Irrespective of molecule, thermal broadening of the absorption features is evident for increasing temperatures. Both effects illustrate the importance of fitting Stark spectra with ground-state absorption spectra recorded at the same temperature.

Figure 4c, d show the change in molar attenuation coefficient $\Delta\epsilon$ scaled to 1 MV/cm for TTF-BTD and anthanthrene as measured by conventional Stark spectroscopy for a relative orientation between probe pulse and THz waveform polarization of 56 deg/61 deg (blue solid curve) and 90 deg (blue dashed curve). Note, that angles smaller than 56 deg are difficult to realize due to above mentioned geometrical constraints. The corresponding results of the THz Stark spectroscopy are shown in Fig. 4e, f. THz results were averaged over a 467-fs time

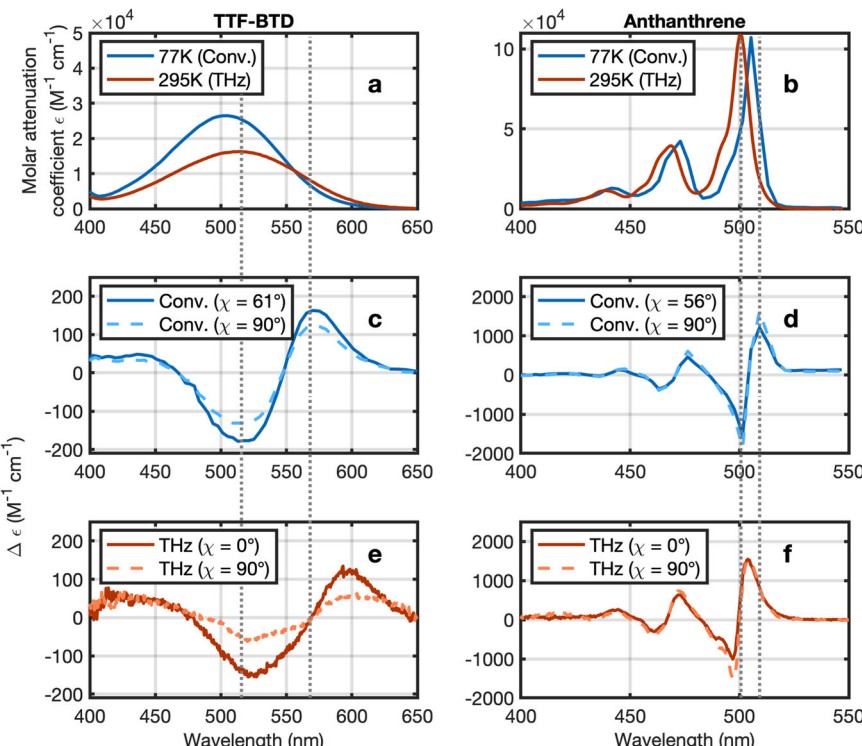

**Fig. 4 | Comparison of conventional and THz Stark spectroscopy. a, b** Low temperature (77 K) and room temperature (295 K) absorption spectra of (**a**) TTF-BTD and (**b**) anthanthrene. **c, d** Conventional Stark spectra measured at 77 K for two values of the angle $\chi$ between the polarization of the THz and the probe pulses. **e, f** THz Stark spectra recorded at room temperature for parallel and perpendicular orientation of THz and probe polarization. For direct comparison the y-scale is in units of $\Delta\epsilon$ scaled to an electric field of 1 MV/cm. The gray dotted vertical lines are guides to the eye and help to visualize the shift of the spectra at the different temperatures.

window around the larger peak of the THz pulse. Unlike conventional Stark spectroscopy, the THz variant has no geometrical constraints and allows for angles down to 0 deg, which helps to improve the dynamic range of measurement as well as the accuracy of the Liptay analysis. Except for the temperature related blue- or red-shift and the different minimal angle, the Stark signatures of both methodologies are in excellent agreement with each other. The peak change in molar attenuation coefficient for TTF-BTD is slightly lower in the THz measurement due to the thermal broadening of the absorption spectrum at room temperature.

### Relevant molecular parameters

A more rigorous comparison becomes available after analyzing the spectra in Fig. 4 following the Liptay protocol. It reveals the change in dipole moment, $\Delta\mu$, the angle between the change in dipole moment and the transition dipole moment $m$, $\zeta$, the average change in polarizability, $\mathrm{Tr}(\Delta\alpha)$, and the change in polarizability parallel to the transition dipole moment $m$, $m\,\Delta\alpha\,m$. The latter consistently showed large fitting errors and is not reported here. Table 1 summarizes these molecular parameters calculated from DFT as well as measured and extracted via the Liptay analysis of both Stark spectroscopy methodologies.

The agreement within the error bars in all molecular parameters extracted from the two Stark spectroscopy modalities is remarkable and confirms that both techniques deliver results that are in quantitative agreement with each other. The biggest source of error comes from the uncertainty of the electric field applied to the sample (Supplementary Information 6). Furthermore, the experimental results are in reasonable agreement with DFT calculations given the fact that the solvent environment can only be treated approximately if at all. Including the solvent effect in the DFT calculations for TTF-BTD leads to marginal modifications to the molecular geometry as compared to the gas phase and details can be found in reference[37]. The mismatch in measured and calculated angle $\zeta$ most likely is due to these structural modifications, but may be also a result of temperature affecting electrostatic interactions and molecular geometry[37,40]. For the anthanthrene, the DFT calculations were performed at 0 K and without solvent effects, nevertheless, the experimental results agree very well with the DFT calculations. Hence, neither the low polarity of the solvent nor the increased temperature seem to drastically affect the dipole moment or the electronic polarizability of the molecule.

In conclusion, we have demonstrated that THz Stark spectroscopy indeed reveals the same physicochemical properties of molecules as conventional Stark spectroscopy, but at the same time opens hitherto inaccessible opportunities, because it is not subjected to the same limitations that apply to conventional Stark spectroscopy. Geometrical constraints are removed allowing for arbitrary angles between probe pulse and THz waveform polarization, no electrodes are required, which helps to avoid potential redox chemistry in the pristine sample, and the much higher frequency of THz waveforms allows for higher electric field strengths before the threshold for dielectric breakdown is reached. Most importantly, however, THz Stark spectroscopy removes the need to immobilize the molecular ensemble by freezing the

solvent. Although this study demonstrates time-resolved THz spectroscopy for two specific molecules in a non-polar environment, the method is not limited to that. As also shown in a recent report on transient THz field-induced broadening of absorption bands, which was published during the reviewing process[41], this opens up interesting future applications to study samples also in polar natural environments. Hence, molecules or bio-molecules can now be studied in their natural environment and at relevant temperatures. Our findings are based on measurements of two molecules relevant in the context of molecular electronics.

In addition, THz Stark spectroscopy allows us to observe transient or non-equilibrium electronic properties of molecules with sub-100 fs resolution. Consequently, THz Stark spectroscopy can be used to study molecular ensembles at conditions not accessible to conventional Stark spectroscopy, for instance, within a much increased range of temperatures or in different non-polar or polar solvents, even those that do not form transparent glasses at low temperatures. Today's high-field THz sources generate field strengths in excess of 1 MV/cm and even higher fields when combined with field enhancement structures so that higher-order Stark contributions become observable, such as hysteresis effects originating from electron-phonon couplings. Higher order Stark contributions are impossible to access via conventional Stark spectroscopy but are relevant to model electron dynamics induced by external or local fields (e.g. charge and electron transfer) or to refine quantum chemistry codes. Moreover, the intrinsic time resolution of around 100 fs facilitates studies on the time-dependent physicochemical properties of a molecule during its photocycle, specifically it allows for Stark spectroscopy of excited states.

## Methods

### THz Stark spectrometer

The THz Stark spectrometer was designed to record the change in absorption $\Delta A(\tau, \lambda) = A_{\mathrm{THz\,on}}(\tau, \lambda) - A_{\mathrm{THz\,off}}(\tau, \lambda)$ as a function of time delay between the THz waveform and the probe pulse $\tau$ and of wavelength $\lambda$. The recorded Stark maps $\Delta A(\tau, \lambda)$ were subsequently corrected for the fs-SC group velocity dispersion and the background resulting from the pure solvent (see Supplementary Information 4). The analysis of the Stark spectra, here at maximum electric field, $\Delta A(\lambda)$, was outlined in reference[5]. After having identified the Stark-active transitions, the Stark spectra were subsequently analyzed with the Liptay formalism, which links the molar absorption $\Delta\epsilon(\bar{\nu})$ as a function of wavenumber to the ground state absorption spectrum $\epsilon(\bar{\nu})$. The model assumes a fixed angle between electric field and probe polarization and an isotropic distribution of transition dipole moments, which is achieved by freezing the sample in conventional Stark spectroscopy. The ground-state absorption spectra and two Stark spectra for different probe polarizations were fitted simultaneously with a weighted sum of the zeroth, first, and second order derivative of the ground state absorption spectrum.

$$\Delta\epsilon(\bar{\nu}) = f_l^2 |\mathbf{E}|^2 \left\{ a\epsilon(\bar{\nu}) + b\frac{\mathrm{d}}{\mathrm{d}\bar{\nu}}\left(\frac{\epsilon(\bar{\nu})}{\nu}\right) + c\frac{\mathrm{d}^2}{\mathrm{d}\bar{\nu}^2}\left(\frac{\epsilon(\bar{\nu})}{\nu}\right) \right\}. \quad (1)$$

From the fit parameters $a$, $b$, and $c$ we extracted the trace of the polarizability tensor, its projection along the transition dipole moment, the angle between the applied electric field and the probe polarization, the change in dipole moment, and the angle between the change in dipole moment and the transition dipole moment. An important ingredient to the fit is the THz electric field strength $|\mathbf{E}|$ in the sample at which the probe pulse interrogates the molecular system. In order to account for all experimental effects we first measured the THz electric field in air and then performed finite difference time-domain simulations to determine the time dependence of the THz electric field experienced by the probe pulse in the complex cuvette/liquid environment. We found that the time dependence is almost

**Table 1 | Comparison of relevant molecular parameters as calculated via DFT or measured by conventional and THz Stark spectroscopy**

| Parameter | DFT | Conventional Stark | THz-Stark |
|---|---|---|---|
| TTF-BTD: tetrathiafulvalene-benzothiadiazole | | | |
| $\Delta\mu$ (D) | 16.2 | 14.7 ± 0.1 | 15.3 ± 1.8 |
| $\zeta$ (°) | ≈ 0 | 24.8 ± 0.2 | 18.7 ± 1.1 |
| Anthanthrene: 4,6,10,12-tetrakis(triisopropylsilylethynyl)anthanthrene | | | |
| $\mathrm{Tr}(\Delta\alpha)$ (Å³) | 457 | 363 ± 20 | 296 ± 70 |

identical, however with the peak electric field strength being reduced by a factor of 0.7 due to Fresnel reflections and Fabry-Perot effects. Hence, the maximum field in the sample was reduced to $(280 \pm 17)$ kV/cm. The electric field experienced by the molecules is further modified by the local field correction factor $f_l$, which is a measure of how the solvent cavity affects the field inside the cavity containing the molecule[42–44]. We estimated the local field correction factors for TTF-BTD and anthanthrene to 1.30 and 1.33 for conventional Stark spectroscopy and to 1.26 and 1.29 respectively for THz Stark spectroscopy (Supplementary Information 8).

## Data availability
All data generated in this study have been deposited in the BORIS repository https://doi.org/10.48620/384. The data supporting the findings of this study are available from the authors upon request.

## Code availability
The Starkfit MATLAB code can also be found on GitHub https://github.com/stanleyextreme/Stark-Spectroscopy-Fitter.

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

## Acknowledgements

We acknowledge experimental support by Steven E. Meckel. This work was supported by the National Center of Competence in Research - Molecular Ultrafast Science and Technology (NCCR MUST), a research instrument of the Swiss National Science Foundation as well as by the Swiss NSF (200021-204053). B.J.K. also acknowledges funding from the European Union's Horizon 2020 research and innovation program under the Marie Skłodowska-Curie grant agreement (FP-RESOMUS - MSCA 801459). R.J.S. acknowledges partial support from the Exobiology program at the National Aeronautics and Space Administration (Grant 80NSSC17K0033). A.B. and M.C. acknowledge the support of the Research Council of Norway through the CoE Hylleraas Centre for Quantum Molecular Sciences (grant no. 262695), and the Norwegian Supercomputing Program NOTUR (grant no. NN4654K).

## Author contributions

T.F. designed and directed this project. B.J.K., E.J.R., D.R. and E.Z. performed the THz-Stark experiments and analyzed the data. B.J.K. and Z.O. built and characterized the high-field THz source. M.A., E.J.R., A.C. and R.J.S. performed the conventional Stark experiments and suggested ways to improve the analysis. G.S., S.D. and S.-X.L. synthesized the molecular samples. A.B. and M.C. performed the DFT calculations. All authors contributed to the discussion, writing, and reviewing of the final manuscript.

## Competing interests

The authors declare no competing interests.
