## [Peer Review File · Nature Communications]

Time-resolved THz Stark spectroscopy in solutionsREVIEWER COMMENTS

Reviewer #1 (Remarks to the Author):

An article that is submitted for publication may be of interest to a terahertz audience. It cannot be said that this is a fairly new application of THz pulsed radiation as a Stark electrical excitation. I would recommend to the second to look at earlier publications on this topic, for example "Toshiki Yamada et al 2019 Jpn. J. Appl. Phys. 58 040901". For those involved in pulsed spectroscopy, it is well known that the "Stark method" is the most accurate for determining the strength of a pulsed THz field. Therefore, the use of the Stark effect in combination with powerful pulsed THz radiation is not a new idea in itself. It was used for solids, for gases and other media. It seems to me that the originality of this work is the use of pulsed THz radiation for the diagnosis of molecular solutions. However, this is also the complexity of the work.

There are a few points worth debating:

1. The article states that the use of a pulsed THz field "Most importantly however, THz Stark spectroscopy removes the need to immobilize molecular ensemble by freezing the solvent". It is assumed that at ω , due to the fact that the molecule does not have time to orient its dipole moment along the electric field vector of the THz pulse, the effect of "freezing" the medium is obtained and the induced anisotropy of the medium can be ignored. This is not the case; a THz pulse propagating in a medium due to the dispersion of the medium, absorption in the THz frequency range rather quickly loses the form of a pulse with one oscillation period. It is necessary to take into account the complex time profile of the THz pulse and its distortion during propagation in the solvent. The article describes well all the possible difficulties of applying the method, but it is not clear how to take this into account. Formula (1) requires additional comments that take into account the real induced anisotropy.

2. The authors of the article used toluene for their experiments. Toluene is a non-polar solvent, but the authors write that "Hence, molecules or bio-molecules can now be studied in their natural environment and at relevant temperatures". For biomolecules "natural environment" is water, which is a polar solvent. Thus, only non-polar solvents can be used. Moreover, it is worth noting that the results of the experiments will depend on the type of solvents.

All this makes the attached method quite difficult to interpret in a wide practical application for the spectroscopy of solutions.

Reviewer #2 (Remarks to the Author):

Bong Joo Kang et al. demonstrate that Stark spectroscopy can also be done with picosecond-long "electric pulses" at a frequency of about 1 THz. They performed time- and polarization-resolved pump-probe experiments wherein the THz pulse is the pump and white-light is the probe, on two well-characterized molecules dissolved in toluene that display either first or second-order Stark effects. Their results agree very well both with traditional Stark spectroscopy results as well as DFT calculations. To my knowledge, this is the first proof that THz-Stark experiments are feasible on molecules in liquid solution, and I share the enthusiasm of the authors for the novel possibilities this technique might open the door to, i.e., the field of molecular Stark spectroscopy at room temperature.

In my opinion the paper is clearly written and the experimental results/analysis are convincing. I have only few comments, as follows:

1) In the abstract, the authors claim that this work allows experiments to be performed at conditions "more relevant to physiological" ones. The introduction states that "freezing prevents molecules to be characterized in their natural, liquid or physiological environment", while the conclusions say "molecules or bio-molecules can now be studied in their natural environment and at relevant temperatures". I think that the authors should revise these statements and be more careful when they discuss natural or physiological conditions of molecules or bio-molecules: life and natural chemistry happens in water, not in toluene. In addition, studying water with intense terahertz spectroscopy is a whole and rich research field in itself, and I have quite a few concerns in extending the THz-Stark spectroscopy used here to the study of molecules in water.

2) What is the dielectric breakdown threshold of Toluene for almost single-cycle THz pulses at ~0.5-1 THz? I suggest that the authors add to the SI a blank, check experiment wherein they measured with the same technique only pure toluene, without molecules dissolved in it. This would quantify precisely the background of their measurements. It would also be nice to see the signal obtained with the empty quartz cuvette, if any.

3) In the abstract, the authors state that "THz pulses oscillate faster than typical molecular rotations". Later on, that "At THz frequencies the electric field oscillates faster than typical molecular rotation times" and "the THz field oscillation is faster than any molecular rotation time". I suggest the authors to revise these statements. While I agree that the rotations of the molecules in solution that they are studying take longer than about 1 ps, this is not true for smaller molecules, i.e., just to make an example, water vapor has a multitude of rotational absorption lines at these frequencies. A related, minor point, regards the THz fields generated by tilted-front optical rectification. Is the optical system purged in nitrogen? Based on the oscillations following the field in Fig.7 in the SI I assume it is not, but it could be appropriate to state this somewhere. (The dip in the spectrum at ~1.1 THz is probably absorption by water vapor?).

4) The "non-Markovian behavior" is stated only in the abstract and in the conclusions. I think that this term can be difficult for the general readership of Nature Communications. I think too much relevance is given to these specific processes (i.e., non-Markovian means that the density functional describing the quantum system cannot be written as a function of one single time variable, but the sequence of light-matter interactions matters). I would suggest to limit the discussion to extreme non-linear / non-perturbative / higher-order Stark contributions.

5) I think there is a typo on line 77, 10^6 V/cm (1 MV/cm)

Reviewer #3 (Remarks to the Author):

The paper "Time-resolved THz Stark" spectroscopy, by Prof Feurer and collaborators is a very interesting and well written report on a highly sensitive spectroscopic method for molecular systems.

The paper is very interesting and very clearly written, the theoretical analysis is appropriate and the technical aspects of the paper are very good, as expected for this group of authors.

I must also say that as a non-expert reader, I have enjoyed reading this paper and I feel I have learned quite a few things.

Having said all the above.

I am still unconvinced that this report, or at least a paper written in this way, should be featured in Nature Communication. Unless I have missed something, the molecular systems chosen by the authors could be studied with the same precision using either conventional Stark spectroscopy AND the newly developed THz Stark spectroscopy, with the caveat that the THz measurements were done at elevated temperatures where Stark spectroscopy is usually less precise.

The authors say that this method could be used to nature transient phenomena, but then they should do this.

If I am allowed an offhand comment, in the hierarchy of features that I, as reviewer, look for in a paper, even before I ask the question of "what did they discover in this physical system and do we care about this discovery?" one should answer the question "have I learned something that I didn't know before?".

I would like to encourage the authors to address the issue more directly. Which features of these two molecular systems have been uncovered? This is regardless of these features are important or not, at least in the first instance.

I maintain an open mind on the manuscript, but I am not satisfied with the current format. After everything is said and done, this paper is too much about the optical technique and reads like an Optics Letter.

I look forward to a more focused discussion along the lines highlighted above. I would like to reiterate that my comments should be taken as constructive criticism.

Response to Reviewer 1

An article that is submitted for publication may be of interest to a terahertz audience. It cannot be said
that this is a fairly new application of THz pulsed radiation as a Stark electrical excitation. I would
recommend to the second to look at earlier publications on this topic, for example "Toshiki Yamada et
al 2019 Jpn. J. Appl. Phys. 58 040901". For those involved in pulsed spectroscopy, it is well known that
the "Stark method" is the most accurate for determining the strength of a pulsed THz field. Therefore,
the use of the Stark effect in combination with powerful pulsed THz radiation is not a new idea in itself.
It was used for solids, for gases and other media. It seems to me that the originality of this work is the
use of pulsed THz radiation for the diagnosis of molecular solutions. However, this is also the complexity
of the work.

Response:

We would like to thank the reviewer for reading the manuscript carefully, for the very valuable comments,
and for pointing out the originality of our work: pulsed THz-induced diagnosis of molecular solutions. We
appreciate the introduction to the successful earlier publications on Stark electrical excitation involved in
pulsed spectroscopy and the way to measure the temporal field strength of the THz field very accurately
by using the first-order Stark effect in nonlinear optical polymers. The material investigated in the paper
was a polymer, which can be measured in the conventional Stark method without freezing. And only one
specific wavelength was used to measure the Stark properties. However, Stark spectroscopy of molecular
solutions at room temperature was considered impossible for decades. Our result proved that time-resolved
55 THz Stark spectroscopy of molecules in solution can be performed without the need to freeze the solvent,
thanks to the recently developed single-cycle THz pulses with sufficiently strong electric fields based on
the phase-stable nature.

Although the main topic is different, methodologically as the reviewer pointed out, it is of a similar
category. Hence, it would be appropriate to introduce previous work related to the Stark effect or the
electro-absorption effect. In the revised version of the manuscript, we cite relevant research papers on the
"Stark method" used for solids, gases, and other media such as quantum dots, semiconductors, polymers
or TMDC. To the best knowledge, our result is the first proof of THz-Stark measurement on molecules
in a liquid solution at room temperature. Recently, the observation of an ultrafast Stark response in
solid materials, where the identical signature is likely to be observed even with the conventional DC-Stark
method, has been published. Moreover, more recent trends reflect how important our research in this field
is.

The list of relevant research papers on Stark method is as follows:

- – B. C. Pein et al. Terahertz-driven luminescence and colossal stark effect in CdSe–CdS colloidal
quantum dots. *Nano Lett.* 17, 5375 (2017).
- – C. Schmidt et al. Signatures of transient Wannier-Stark localization in bulk gallium arsenide. *Nature*
*Communications* 9, 2890 (2018).
- – T. Yamada et al. Terahertz wave detection by the Stark effect in nonlinear optical polymers. *Japanese*
*Journal of Applied Physics* 58, 040901 (2019).
- – B. C. Pein et al. Terahertz-Driven Stark Spectroscopy of CdSe and CdSe-CdS Core-Shell Quantum
Dots. *Nano Lett.* 19, 8125 (2019).
- – T. Barmashova et al. Studies of terahertz discharge in noble gases using a Michelson interferometer.
*J. Phys.: Conf. Ser.* 1697 012220 (2020).
- – J. Shi et al. Room Temperature Terahertz Electroabsorption Modulation by Excitons in Monolayer
Transition Metal Dichalcogenides. *Nano Lett.* 20, 5214 (2020).
- – T. Venanzi et al. Terahertz-Induced Energy Transfer from Hot Carriers to Trions in a MoSe₂ Mono-
layer. *ACS Photonics* 8, 2931 (2021).
- – C. Gollner et al. Ultrafast Electro-Absorption Switching in Colloidal CdSe/CdS Core/Shell Quantum
Dots Driven by Intense THz Pulses. *Adv. Optical Mater.* 10, 2102407 (2022).
- – M. B. Heindl et al. Ultrafast imaging of terahertz electric waveforms using quantum dots. *Light:*
*Science & Applications* 11, 5 (2022).
- – Y. Kobayashi et al. Floquet engineering of strongly driven excitons in monolayer tungsten disulfide.
*Nature Physics* 19, 171(2023).

Action:

1. In the abstract, the introduction and the conclusion section, we now better explain the focus of
our work, our key results, their novelty and their relevance. We added text to the introduction
summarizing recent trends in related THz studies (end of 1st Paragraph):

“In addition, more recent research results revived the interest in the observation of transient Stark
responses in solids, gases, or other media such as polymers, quantum dots, and TMDC [ref]. How-
ever, transient Stark spectroscopy of molecules in solution at room temperature has not yet been
demonstrated, because a successful measurement of molecules in solution is subject to some pre-
conditions.”

2. In addition, to emphasize the originality of our work and in order to accurately reflect the content of
our research, the title of the manuscript has been changed to: ‘Time-resolved THz Stark spectroscopy
of molecules in solution’.

There are a few points worth debating:

- 1) The article states that the use of a pulsed THz field “Most importantly however, THz Stark spectroscopy
removes the need to immobilize molecular ensemble by freezing the solvent”. It is assumed that at, due to
the fact that the molecule does not have time to orient its dipole moment along the electric field vector of
the THz pulse, the effect of “freezing” the medium is obtained and the induced anisotropy of the medium
can be ignored. This is not the case; a THz pulse propagating in a medium due to the dispersion of
the medium, absorption in the THz frequency range rather quickly loses the form of a pulse with one
oscillation period. It is necessary to take into account the complex time profile of the THz pulse and
its distortion during propagation in the solvent. The article describes well all the possible difficulties of
applying the method, but it is not clear how to take this into account. Formula (1) requires additional
comments that take into account the real induced anisotropy.

**Response:**

Thank you for pointing out this critical issue and we apologize for misleading or missing explanations. We
would like to emphasize that the sample molecules dissolved in toluene passed from a reservoir through
a flow cell with an inner thickness of 200 μm and two 200 μm thick fused silica windows on both sides.
The THz power transmission through the sample cell was measured to be more than 90%. This is because
(1) the sample is very thin, (2) toluene being a non-polar solvent absorbs very little THz radiation [Ref
and measurement], (3) the THz absorbance of the molecules is negligible [measurement] since there is no
molecular resonance within or close to the THz pulse spectrum. Moreover, the refractive indices of fused
silica as well as toluene are rather flat in the frequency range between 0.3 THz and 1.5 THz, which is why
our experimental conditions allow us to neglect dispersion and absorption, and consequently any distortion
of the THz waveform as it propagates through the cuvette containing the sample. On more quantitative
grounds, we performed Finite Difference Time Domain simulations of the THz pulse propagating through
the cuvette taking into account absorption and dispersion of the windows as well as the solvent. As a result
we could determine the actual electric field experienced by the broadband probe pulse. These simulations
showed no evidence of reshaping of the THz waveform. Further experimental evidence comes from the
fact that the Stark signature for a selected wavelength as a function of time delay follows the square of the
electric field as measured by electro-optic sampling. Any distortion would manifest itself in a noticeable
deviation.

While the experimental results show that sample molecules do not rotate as a result of the applied THz
electric field (absence of a zero order Stark signature), the solvent molecules might still be affected resulting
in a THz-induced anisotropy, which in turn might modify the Stark-signature of the sample. In this case,
Equation (1) would indeed require modifications to take into account this induced anisotropy. Yet, we
find that this is of no concern because toluene is a non-polar solvent and does not — to first order —
interact with the THz electric field as it would be the case for a polar solvent, such as water. This is also
confirmed by measurements with pure toluene.

- Ref: M. Sajadi et al. Transient birefringence of liquids induced by terahertz electric-field torque on
permanent molecular dipoles. Nat. Commun. 8, 14963 (2017).

In addition, in Equation (1), by identifying the Stark-active transitions, we link the signal to the molar
absorption. Specifically, the model requires two fixed angles between the electric field and probe po-
larization and an isotropic distribution of transition dipole moments. The two Stark spectra are fitted
simultaneously with a weighted sum of the zeroth, first and second-order derivative of the ground-state
absorption spectrum. We observed a negligible background signal from the solvent, which might result
from a small change in the ground state absorption of the solvent (toluene [Ref1] and EtOAc [Ref2]) in
the SC-spectral range. Note, that the background signal measured for pure solvent, which turns out to

be unrelated to the Stark-signature, is subtracted from all measurements prior to the fitting procedure
described above.

- Ref1: Meng Wang et al., A Demonstration of Broadband Cavity-Enhanced Absorption Spectroscopy at
Deep-Ultraviolet Wavelengths: Application to Sensitive Real-Time Detection of the Aromatic Pollutants
Benzene, Toluene, and Xylene, *Anal. Chem.* 94, 10, 4286–4293 (2022).

- Ref2: B. Benayada et al., UV absorption cross sections for acetates, *Desalination*, 126, 83 (1999).

- 2) The authors of the article used toluene for their experiments. Toluene is a non-polar solvent, but the
authors write that "Hence, molecules or bio-molecules can now be studied in their natural environment
and at relevant temperatures". For biomolecules "natural environment" is water, which is a polar solvent.
Thus, only non-polar solvents can be used. Moreover, it is worth noting that the results of the experiments
will depend on the type of solvents. All this makes the attached method quite difficult to interpret in a
wide practical application for the spectroscopy of solutions.

**Response:**

Thank you for pointing out this potentially exaggerated statement in our paper. With the molecules
presented in this publication we cannot fully support this claim, because they don't dissolve well in polar
solvents. However, we have included measurements in the solvent with the highest possible polarity while
still being able to reach a sufficiently high molar concentration. In fact, in our next project is to investigate
molecular Stark effect in polar solvents, such as water, but with different molecules. Note that the primary
goal of this project was (1) to demonstrate transient THz Stark measurements of molecules in solvent
without freezing, (2) to compare the THz Stark method at room temperature with conventional Stark
spectroscopy at low temperatures, which can only be performed with a few selected solvents including
toluene, and (3) to elucidate the intrinsic properties of the molecules themselves.

In principle, time-resolved Stark spectroscopy would allow for resolving effects related to a rearrangement
of solvent molecules. Hence, it is necessary to understand the signatures seen depending on the type of
solvent, as the reviewer pointed out. As a consequence, THz Stark spectroscopy can provide interesting
new insights into the nature of the interactions between solute molecules and solvent [Ref1, Ref2].

– Molecules in non-polar solvent: In non-polar solvents, the Stark effect provides a direct measurement
of changes in the molecule's electronic structure or its electronic transition energies induced by the
applied electric field.

– Molecules in polar solvent: In polar solvents, the applied electric field will interact with the solvent
molecules as well, which may result in structural modifications. This can affect the local environment
around a solute molecule influencing, for instance, its ro-vibrational or electronic spectra.

- [Ref1] Gerold U. Bublitz et al., Stark Spectroscopy of Donor/Acceptor Substituted Polyenes. *J. Am.*
*Chem. Soc.* 119, 3365-3376 (1997)

- [Ref2] K. Nauta et al., Stark Spectroscopy of Polar Molecules Solvated in Liquid Helium Droplets. *Phys.*
*Rev. Lett.* 82, 4480 (1999)

In polar solvents we need to consider different Stark responses and may — to first order — express the
overall response as a sum of different contributions:

$$R = R_{\text{solute}} + R_{\text{solvent}} + R_{\text{solute-solvent}}$$

Note that in non-polar solvents, R_{solvent} , $R_{\text{solute-solvent}}$ are negligible. Moreover, a non-negligible R_{solute}
is a prerequisite to examine solute-solvent interactions.

In order to at least partially substantiate our claim, the same solute molecule was measured in ethyl
acetate (EtOAc) with a relative polarity of about 0.228. Its relative polarity is about 2.3 times higher
than that of toluene (relative polarity 0.099). After an extensive search we found that EtOAc is the solvent
with the highest polarity in which the solute molecules would dissolve in a sufficiently high concentration.
The first observation to note is a blue-shift of the ground state absorption spectrum, which we attribute
to the solvent's relative polarity. Except for this shift, the shape of the Stark signatures in EtOAc are
almost identical to those measured in toluene.

Moreover, the molecular electronic properties that result from the analysis of the Stark spectra measured
in the two solvents are the same within the errors (see table 1). Hence, we believe the statement that

Figure 1: Stark signal comparison for Anthanthrene in toluene and EtOAc

Table 1: Comparison of relevant molecular parameters as calculated via DFT or measured by conventional and THz Stark spectroscopy.

Parameter	DFT	Conventional Stark (Toluene)	THz-Stark (Toluene)	THz-Stark (EtOAc)
anthanthrene: 4,6,10,12-tetrakis(triisopropylsilylethynyl)anthanthrene				
$\text{Tr}(\Delta\alpha)$ (\AA^3)	457	363 ± 20	296 ± 70	229 ± 53

molecules or bio-molecules can be studied in their natural environment and at relevant temperatures is not completely wrong. Our future studies will shed more light on this question.

Action:

1. Abstract: We have rephrased the entire abstract. Particularly we modified the sentence, "Hence, THz Stark spectroscopy allows for time-resolved studies at arbitrary temperatures, specifically ambient conditions more relevant to physiological or operative conditions." to "Consequently, intense THz fields enable dynamic Stark spectroscopy with sub-picosecond time resolution in solution at arbitrary temperatures. We find that this method is applicable in both non-polar and polar solvents, opening the way to physiological or operational conditions in the future. Moreover, dynamical field effects, e.g., higher-order Stark contributions or hysteresis effects, can be studied on the time scales of molecular vibrations or rotations."
2. Conclusion: We have added the sentence, as suggested, in brackets. "... removes the need to immobilize the molecular ensemble by freezing the solvent. Although this study demonstrates time-resolved THz spectroscopy for two specific molecules in a non-polar environment, the method is not limited to that. This opens up interesting future applications to study samples also in polar natural environments."
3. Supplementary Information: We have added the additional experimental results of Anthanthrene in EtOAc and some explanatory text.

Response to Reviewer 2

Bong Joo Kang et al. demonstrate that Stark spectroscopy can also be done with picosecond-long "electric pulses" at a frequency of about 1 THz. They performed time- and polarization-resolved pump-probe experiments wherein the THz pulse is the pump and white-light is the probe, on two well-characterized molecules dissolved in toluene that display either first or second-order Stark effects. Their results agree very well both with traditional Stark spectroscopy results as well as DFT calculations. To my knowledge, this is the first proof that THz-Stark experiments are feasible on molecules in liquid solution, and I share the enthusiasm of the authors for the novel possibilities this technique might open the door to, i.e., the field of molecular Stark spectroscopy at room temperature.

Response:

We are honored by the reviewer's acknowledgment of the pioneering nature of our work. The potential pathways this technique opens are truly exciting, and we appreciate the shared enthusiasm regarding the prospective opportunities it presents.

In my opinion the paper is clearly written and the experimental results/analysis are convincing. I have only few comments, as follows:

- 1) In the abstract, the authors claim that this work allows experiments to be performed at conditions "more relevant to physiological" ones. The introduction states that "freezing prevents molecules to be characterized in their natural, liquid or physiological environment", while the conclusions say "molecules or bio-molecules can now be studied in their natural environment and at relevant temperatures". I think that the authors should revise these statements and be more careful when they discuss natural or physiological conditions of molecules or bio-molecules: life and natural chemistry happens in water, not in toluene. In addition, studying water with intense terahertz spectroscopy is a whole and rich research field in itself, and I have quite a few concerns in extending the THz-Stark spectroscopy used here to the study of molecules in water.

Response:

We thank the reviewer for these comments. It was our intention to highlight the potential application of THz Stark spectroscopy for natural environments at relevant ambient temperatures. We totally agree with the reviewer that there are concerns about the possibility of extending THz Stark spectroscopy to polar solvents specifically, water. In polar solvents, the solvent molecules interact with the applied electric field, which in turn will influence the solute-solvent interaction. We have tried to substantiate our claim by measuring the Stark spectra of Anthanthrene in EtOAc, the solvent with the highest polarity that would dissolve the solute. Of course, EtOAc is not as polar as water, but has a 2.3 times higher polarity compared to toluene.

Action:

We revised the manuscript including Abstract, Introduction and Conclusions according to the reviewer's recommendation. We now mention polar solvents only in the Outlook section to highlight the potential application of THz Stark spectroscopy for future research.

1. Abstract: As suggested by the reviewer, we have rephrased the entire abstract. Particularly we modified the sentence, "Hence, THz Stark spectroscopy allows for time-resolved studies at arbitrary temperatures, specifically ambient conditions more relevant to physiological or operative conditions." to "Consequently, intense THz fields enable dynamic Stark spectroscopy with sub-picosecond time resolution in solution at arbitrary temperatures. We find that this method is applicable in both non-polar and polar solvents, opening the way to physiological or operational conditions in the future. Moreover, dynamical field effects, e.g., higher-order Stark contributions or hysteresis effects, can be studied on the time scales of molecular vibrations or rotations."
 2. Conclusion: We have added the sentence, as suggested: "... removes the need to immobilize the molecular ensemble by freezing the solvent. Although this study demonstrates time-resolved THz spectroscopy for two specific molecules in a non-polar environment, the method is not limited to that. This opens up interesting future applications to study samples also in polar natural environments."
- 2) What is the dielectric breakdown threshold of Toluene for almost single-cycle THz pulses at 0.5-1 THz? I suggest that the authors add to the SI a blank, check experiment wherein they measured with the same technique only pure toluene, without molecules dissolved in it. This would quantify precisely the

background of their measurements. It would also be nice to see the signal obtained with the empty quartz cuvette, if any.

Response:

Apologies for not providing more comprehensive information about our measurement on pure toluene. From literature it is known that the dielectric breakdown voltage of toluene is around 250 kV/cm at DC [Ref1]. But the threshold is expected to be at least a factor of 10 higher at THz frequency due to its frequency-dependent scaling properties [Ref2]. Therefore, we can safely neglect this in our considerations. In addition, please find the Stark measurements of pure toluene in the supplementary information (SI). Note, in all measurements reported we have subtracted the data recorded (background) for pure toluene.

- [Ref1] Y. Nitta et al., Polarity Effect on Breakdown Voltage in Organic Liquids. IEEE Transactions on Electrical Insulation, EI-11(3), 91-94 (1976).
- [Ref2] J. W. McPherson et al., Physical model for the frequency dependence of time-dependent dielectric breakdown (TDDDB). AIP Advances, 13(5), 055217 (2023).

Action:

1. In response to the reviewer's suggestion, we have added an additional clarification in Section 2.1 "Dynamics of the Stark signature" following the sentence "Calculating the margin along the delay axis between the two green dotted vertical lines results in the average Stark signal versus wavelength (green curve in Fig. 2b)." The added text reads: "Note that all measurements had the background signal (from pure toluene in the cuvette) subtracted."
- 3) In the abstract, the authors state that "THz pulses oscillate faster than typical molecular rotations". Later on, that "At THz frequencies the electric field oscillates faster than typical molecular rotation times" and "the THz field oscillation is faster than any molecular rotation time". I suggest the authors to revise these statements. While I agree that the rotations of the molecules in solution that they are studying take longer than about 1 ps, this is not true for smaller molecules, i.e., just to make an example, water vapor has a multitude of rotational absorption lines at these frequencies. A related, minor point, regards the THz fields generated by tilted-front optical rectification. Is the optical system purged in nitrogen? Based on the oscillations following the field in Fig.7 in the SI I assume it is not, but it could be appropriate to state this somewhere. (The dip in the spectrum at 1.1 THz is probably absorption by water vapor?).

Response:

We acknowledge that our statement may not be entirely correct for the smallest molecules. However, most molecules in solutions show rotation times on the order of tens of ps [Ref1-5] and therefore rotate much slower than our 0.6 ps (FWHM) THz pulse. Thus we believe our statement is correct, but to clarify we specified molecular rotation in solutions.

In response to the second part of the comment, we modified the text to make it clear that the optical system is purged neither by dry air nor nitrogen. Therefore, the observed dip in the spectrum at around 1.1 THz is attributed to water vapor absorption.

- [Ref1] R. Damari et al., Coherent radiative decay of molecular rotations: a comparative study of terahertz-oriented versus optically aligned molecular ensembles. Phys. Rev. Lett. 119, 033002 (2017).
- [Ref2] S. Yuan et al., Pulse polarization evolution and control in the wake of molecular alignment inside a filament. Optics Express 23(5), 5582 (2015).
- [Ref3] S. Fleischer et al., Molecular Orientation and Alignment by Intense Single-Cycle THz Pulses. Phys. Rev. Lett. 107, 163603 (2011).
- [Ref4] J. Lu et al., Nonlinear two-dimensional terahertz photon echo and rotational spectroscopy in the gas phase. PNAS 113 (42) 11800 (2016).
- [Ref5] [Book] N.-G. Park, M. Grätzel, T. Miyasaka. Organic-inorganic halide perovskite photovoltaics. 1st ed. (Springer, Berlin, 2016) page 14.

Action:

To make it more clear, we have revised the statements related to molecular rotation time.

1. Abstract: "THz pulses oscillate faster than typical molecular rotations" to "The THz-driven Stark effect is shown to instantaneously trace the applied electric field, proving that solute molecules do not have time to orient their dipole moments, because THz fields oscillate faster than typical molecular rotation times in solution."
2. Introduction: "At THz frequencies the electric field oscillates faster than typical molecular rotation times" to "At THz frequencies the electric field oscillates faster than typical molecular rotation times in solution".

3. Section 2.1: Dynamics of the Stark signature: "the THz field oscillation is faster than any molecular
rotation time" to "the THz field oscillation is faster than the rotation time of the investigated
molecules in solution."
4. Additionally, the water vapor absorption is now mentioned in Section 3.5, "Characterization of THz
pulses", in the SI. "The corresponding THz spectral amplitude shows a center frequency of 0.5 THz
with a bandwidth of 0.7 THz (FWHM). The optical system was in ambient atmosphere, hence the
absorption feature around 1.1 THz is due to water vapor absorption."

4) The "non-Markovian behavior" is stated only in the abstract and in the conclusions. I think that this term
can be difficult for the general readership of Nature Communications. I think too much relevance is given
to these specific processes (i.e., non-Markovian means that the density functional describing the quantum
system cannot be written as a function of one single time variable, but the sequence of light-matter
interactions matters). I would suggest to limit the discussion to extreme non-linear / non-perturbative /
higher-order Stark contributions.

**Response:**

We appreciate the reviewer's thoughtful feedback. We modified the text after taking into account the
possible difficulties this may provide for the general readership of Nature Communications.

**Action:**

We have removed "non-Markovian behavior" in our manuscript as suggested by the reviewer.

5) I think there is a typo on line 77, 10^6 V/cm (1 MV/cm)

**Response:**

We apologize for the typo.

**Action:**

We have corrected the typo and we checked all related values.

Response to Reviewer 3

The paper "Time-resolved THz Stark" spectroscopy, by Prof Feurer and collaborators is a very interesting and well written report on a highly sensitive spectroscopic method for molecular systems. The paper is very interesting and very clearly written, the theoretical analysis is appropriate and the technical aspects of the paper are very good, as expected for this group of authors. I must also say that as a non-expert reader, I have enjoyed reading this paper and I feel I have learned quite a few things.

Response: We are pleased that the reviewer appreciated our work, and we would like to express our gratitude for their interest, as well as their detailed and constructive feedback on our project.

Having said all the above. I am still unconvinced that this report, or at least a paper written in this way, should be featured in Nature Communication. Unless I have missed something, the molecular systems chosen by the authors could be studied with the same precision using either conventional Stark spectroscopy AND the newly developed THz Stark spectroscopy, with the caveat that the THz measurements were done at elevated temperatures where Stark spectroscopy is usually less precise. The authors say that this method could be used to nature transient phenomena, but then they should do this. If I am allowed an offhand comment, in the hierarchy of features that I, as reviewer, look for in a paper, even before I ask the question of "what did they discover in this physical system and do we care about this discovery?" one should answer the question "have I learned something that I didn't know before?". I would like to encourage the authors to address the issue more directly. Which features of these two molecular systems have been uncovered? This is regardless of these features are important or not, at least in the first instance. I maintain an open mind on the manuscript, but I am not satisfied with the current format. After everything is said and done, this paper is too much about the optical technique and reads like an Optics Letter. I look forward to a more focused discussion along the lines highlighted above. I would like to reiterate that my comments should be taken as constructive criticism.

Response:

We thank the reviewer for reading the manuscript carefully and for the very valuable comments to improve our manuscript. We would like to emphasise that for decades it was considered impossible to perform Stark spectroscopy at room temperature, that is without freezing the sample in order to immobilize the solute molecules. With our contribution we have shown that this limitation no longer applies and we consider this a major step forward in Stark spectroscopy. Our results proof that the field-induced Stark effect instantaneously and quadratically traces the applied electric field with sub-ps modulations for both linear and quadratic Stark effect. Negligible memory effects, molecular alignment and influence of the polarity of the THz electric field were observed; our work has proven that the solute molecules do not have time to orient their dipole moments along the electric field vector of the THz pulse, because this oscillates too fast. This is an important experimental finding as well as it allows us to use the very same formalism to extract molecular electronic properties that was developed for conventional Stark spectroscopy. In the revised version we have also included experimental results for a mildly polar solvent (all solvents with even higher polarity do not allow to dissolve a sufficient amount of solute molecules) to at least indicate that the method presented will very likely also work for polar solvents, such as water, which would open entire new avenues for Stark spectroscopy of bio-molecules in their natural environment. Given the nature of the experiment, we are now able to explore new opportunities at room temperature that were inaccessible before, for instance, the interaction between solvent and solute at different temperatures and concentrations, which is impossible in the conventional Stark spectroscopy. The sub-ps time resolution allows to observe time-dependent physicochemical properties of a molecule during, for instance, its photocycle upon optical excitation. In addition, the method allows investigating the dynamics of molecular alignment and reorientation. Hence, our result represents the first stepping stone in unlocking a variety of previously inaccessible Stark spectroscopies.

Action:

In order to address the reviewer's comment "what did they discover in this physical system and do we care about this discovery?" and the question "have I learned something that I didn't know before?" we rephrased the entire abstract:

For decades, it was considered all but impossible to perform Stark spectroscopy on molecules in liquid solution, because their concomitant orientation to the applied electric field results in an overwhelming background signal. A way out was to immobilize the solute molecules by freezing the solvent at sufficiently

low temperatures. While mitigating solute orientation, freezing removes the possibility to study molecules
in liquid environments at ambient conditions. Here we demonstrate time-resolved THz Stark spectroscopy,
which uses intense single-cycle terahertz pulses as electric field source. The THz-driven Stark effect is
shown to instantaneously trace the applied electric field, proving that solute molecules do not have time
to orient their dipole moments, because THz fields oscillate faster than typical molecular rotation times in
solution. Consequently, intense THz fields enable dynamic Stark spectroscopy with sub-picosecond time
resolution in solution at arbitrary temperatures. We find that this method is applicable in both non-polar
and polar solvents, opening the way to physiological or operational conditions in the future. Moreover,
dynamical field effects, e.g., higher-order Stark contributions or hysteresis effects, can be studied on the
time scales of molecular vibrations or rotations. We verify THz Stark spectroscopy for two judiciously
selected molecular systems and compare the results to conventional Stark spectroscopy and first principle
calculations.

REVIEWER COMMENTS

Reviewer #1 (Remarks to the Author):

I read the article carefully and the comments to the reviewers' comments. I cannot say that the article has become more suitable for publication in the magazine Nature Communication& The relevance of the proposed method has not been proven, and its information content is quite limited. The article does not provide the main values of sensitivity, time resolution and other characteristics of the validation of the method. I can agree with one of the reviewers that this is a good article for Optics Letters.

It is very important that there is no comparison of the proposed method with already well-established methods of time-resolved electron spectroscopy.

Reviewer #2 (Remarks to the Author):

I believe that the Authors have convincingly addressed all the points raised and would recommend publication of this work.

However, I discovered very recently this paper by the Elsaesser group: "Transient Terahertz Stark Effect: A Dynamic Probe of Electric Interactions in Polar Liquids" J. Phys. Chem. Lett. 2023, 14, 24, 5505–5510

I am sorry to say this, but I think that the Authors should now include a thorough comparison of their work with this recent one and prove which unique, novel points are still there to warrant publication in Nature Communications.

Reviewer #3 (Remarks to the Author):

In the aggregate, I am satisfied with the revised version of the manuscript. I realise that my request of demonstrating this technique by solving a genuine new problem in physics or chemistry may be asking too much.

I feel that with these changes the manuscript is clearer and I support publication of this as a "technique paper". The manuscript has several notable features, including its clarity, the quality of data and the extensive literature review.

I think that it is also a paper that speaks to laser spectroscopists as well as to molecular chemists, and this is another nice feature of the paper, which makes it suitable for Nature Communications.

Response to Reviewer 1

I read the article carefully and the comments to the reviewers' comments. I cannot say that the article has become more suitable for publication in the magazine Nature Communication & The relevance of the proposed method has not been proven, and its information content is quite limited. The article does not provide the main values of sensitivity, time resolution and other characteristics of the validation of the method. I can agree with one of the reviewers that this is a good article for Optics Letters. It is very important that there is no comparison of the proposed method with already well-established methods of time-resolved electron spectroscopy.

Response:

We would like to thank the reviewer for further valuable comments and for pointing out missing information.

We have revised the manuscript and have included the missing information on sensitivity and time resolution. We believe that we have provided sufficient characteristics to validate the method, i.e. by analyzing margins, by comparing the results to quantum calculations, or by comparing the results including Stark signal and extracted molecular parameters with standard Stark spectroscopy measurements. Note that our method can be applied without freezing at room temperature, which was considered impossible for decades.

Indeed photo-electron spectroscopy has developed into an exciting new tool for liquid phase spectroscopy, to call it well-established is — from our perspective — somewhat far-fetched. During the past few weeks, we have contacted experts in the field and have searched through the literature with the result that we can confirm the reviewer inasmuch as photo-electron spectroscopy could indeed provide very similar information. However, experts have also confirmed that such experiments would be very demanding and to the present day have neither been performed nor published.

Action:

1. Regarding “Sensitivity and time resolution”,

Sensitivity: It is not easy to accurately determine the sensitivity, because the stability of such broadband spectrum is not the same over the entire spectral region, which results in the signal-to-noise to depend on wavelength. Based on our measurement of solvents only, the wavelength averaged sensitivity is determined to be approximately 20 μ OD. Note that we reported the sensitivity “as well as a noise level of 20 μ OD due to fs-SC fluctuations” in ‘Section 2.1 Dynamics of the Stark signature’. However, to indicate it more clearly, “noise level” is changed to “sensitivity level”.

Time resolution: The full width at half maximum of the single-cycle THz pulse is approximately 0.6 ps with a carrier frequency of 0.6 THz. The time resolution of the Stark measurement is, very similar to the electro-optic sampling measurement, determined by the probe pulse duration of approximately 100 fs. To allow for a slight oversampling we chose the time delay increment to 66.7 fs corresponding to 10 μ m steps of the motorized delay stage, which is the time interval in all figures shown. We modified the text in section 2.1 Dynamics of the Stark signature to include the time resolution of our system: The two-dimensional distribution reveals both temporal and spectral characteristics of the THz field-induced Stark shift with a time and wavelength resolution of approximately 100 fs and 0.5 nm.

Response to Reviewer 2

I believe that the Authors have convincingly addressed all the points raised and would recommend publication of this work.

However, I discovered very recently this paper by the Elsaesser group: "Transient Terahertz Stark Effect: A Dynamic Probe of Electric Interactions in Polar Liquids" *J. Phys. Chem. Lett.* 2023, 14, 24, 5505–5510

I am sorry to say this, but I think that the Authors should now include a thorough comparison of their work with this recent one and prove which unique, novel points are still there to warrant publication in *Nature Communications*.

Response:

We are happy to read that we have been able to convincingly address all comments. As requested by the reviewer and after coordinating with the editor we now include a short paragraph on the publication by Singh et al., which appeared during the review process.

In this publication the authors aim at quantifying the electric forces acting on molecules in solution at ambient temperatures with the aim to reveal electronic and optical properties. They use transient THz Stark spectroscopy to modify the electronic absorption of a dye molecule and determine the underlying molecular interactions and dynamics. They report on transient THz field-induced broadening of absorption bands with a small influence of the solvent dynamics. In contrast to our experiments, they adopted a field confinement structure to enhance the transient THz electric field, which helps to observe higher-order Stark contributions that we mention in the conclusion section. Even though the concept of the two publications seems to be similar, we believe that our work still has unique and novel contributions as follows:

1. It is the first report on THz field-induced Stark effect of non-polar molecules in solution without freezing.
2. In order to unambiguously demonstrate that not immobilizing the sample by freezing results in the correct electronic parameters, we included a direct comparison between THz Stark spectroscopy at room temperature and conventional Stark spectroscopy at 77 K.
3. Two distinct molecules were investigated to separately demonstrate the method for linear and quadratic Stark effect.
4. To extract molecular parameters the well-established Liptay analysis was applied by measuring two orientations of THz and probe polarization.
5. For comparison and validation, quantum calculations were performed.

Action:

1. We added text to the conclusion including recent contribution in this field. In the second paragraph of the conclusion: Although this study demonstrates time-resolved THz spectroscopy for two specific molecules in a non-polar environment, the method is not limited to that. As also shown in a recent report on transient THz field-induced broadening of absorption bands, which was published during the reviewing process, [*J. Phys. Chem. Lett.* 2023, 14, 5505-5510] this opens up interesting future applications to study samples also in polar natural environments. In the second paragraph of the conclusion: Today's high-field THz sources generate field strengths in excess of 1 MV/cm and even higher fields when combined with field enhancement structures so that higher-order Stark contributions become observable, such as hysteresis effects originating from electron-phonon couplings. Higher order Stark contributions are impossible to access via conventional Stark spectroscopy but are relevant to model electron dynamics induced by external or local fields (e.g. charge and electron transfer) or to refine quantum chemistry codes. Moreover, the intrinsic time resolution of around 100 fs facilitates studies on the time-dependent physicochemical properties of a molecule during its photo-cycle, specifically it allows for Stark spectroscopy of excited states.

Response to Reviewer 3

In the aggregate, I am satisfied with the revised version of the manuscript. I realise that my request of demonstrating this technique by solving a genuine new problem in physics or chemistry may be asking too much.

I feel that with these changes the manuscript is clearer and I support publication of this as a "technique paper". The manuscript has several notable features, including its clarity, the quality of data and the extensive literature review. I think that it is also a paper that speaks to laser spectroscopists as well as to molecular chemists, and this is another nice feature of the paper, which makes it suitable for Nature Communications.

Response:

We are honored by the reviewer's acknowledgment of the pioneering nature of our work. The potential pathways this technique opens are truly exciting, and we appreciate the shared enthusiasm regarding the prospective opportunities it presents. We are happy to read that our contribution in its revised version is suitable for publication in Nature Communications.

REVIEWERS' COMMENTS

Reviewer #1 (Remarks to the Author):

The authors of the article prepared good answers to the reviewers, which were partially transferred into the text of the article. I think that the article in this form can be published.

Reviewer #2 (Remarks to the Author):

I think that the Authors have addressed all my concerns and recommend publication.

There is only one final small thing: I beg the Authors to remove the term "physiological" from the abstract. Physiological conditions mean with water, which was not measure here, and for which I expect a more complex nonlinear response that cannot be simply extended from the current results. By insisting on using this term they might inadvertently alienate some colleagues who actually work with strong THz on water.

Response to Reviewer 1

The authors of the article prepared good answers to the reviewers, which were partially transferred into the text of the article. I think that the article in this form can be published.

Response: We sincerely thank you for recognizing our efforts. We put a lot of thought into transferring the reviewer's comments into the text of the article as much as possible. We are very pleased and share the enthusiasm on the possibilities of our proposed method, which opens the door to molecular Stark spectroscopy at room temperature.

Response to Reviewer 2

I think that the Authors have addressed all my concerns and recommend publication.

There is only one final small thing: I beg the Authors to remove the term "physiological" from the abstract. Physiological conditions mean with water, which was not measure here, and for which I expect a more complex nonlinear response that cannot be simply extended from the current results. By insisting on using this term they might inadvertently alienate some colleagues who actually work with strong THz on water.

Response: We are delighted to address all reviewer's concerns and are honored by the reviewer's recommendation for publication. We wanted to emphasize the potential pathways of this technique, opening a truly exciting research field. We recognized that it could sometimes unintentionally alienate some of our colleagues actually working in the potential research field.

Action:

1. We removed the term "physiological" from the abstract as the reviewer requested.